# MULTI-REPRESENTATION ATTENTION FRAMEWORK FOR UNDERWATER BIOACOUSTIC DENOISING AND RECOGNITION

## ABSTRACT

Automated monitoring of marine mammals in the St. Lawrence Estuary faces extreme challenges: calls span low-frequency moans to ultrasonic clicks, often overlap, and are embedded in variable anthropogenic and environmental noise. We introduce a multi-step, attention-guided framework that *first* segments spectrograms to generate soft masks of biologically relevant energy and *then* fuses these masks with the raw inputs for multi-band, denoised classification. Image and mask embeddings are integrated via mid-level fusion, enabling the model to focus on salient spectrogram regions while preserving global context. Using real-world recordings from the Saguenay–St. Lawrence Marine Park Research Station in Canada, we demonstrate that segmentation-driven attention and mid-level fusion improve signal discrimination, reduce false positive detections, and produce reliable representations for operational marine mammal monitoring across diverse environmental conditions and signal-to-noise ratios. Beyond in-distribution evaluation, we further assess the generalization of Mask-Guided Classification (**MGC**) under distributional shifts by testing on spectrograms generated with alternative acoustic transformations. While high-capacity baseline models lose accuracy in this Out-of-distribution (OOD) setting, MGC maintains stable performance, with even simple fusion mechanisms (gated, concat) achieving comparable results across distributions. This robustness highlights the capacity of MGC to learn transferable representations rather than overfitting to a specific transformation, thereby reinforcing its suitability for large-scale, real-world biodiversity monitoring. We show that in all experimental settings, the MGC framework consistently outperforms baseline architectures, yielding substantial gains in accuracy on both in-distribution and OOD data.

## 1 INTRODUCTION

The St. Lawrence Estuary is an acoustic habitat where protected marine mammal species must maintain essential biological functions, communication, navigation, and foraging, in the presence of increasing anthropogenic noise. Ship noise can mask calls and echolocation, disrupt essential behavioral sequences, and induce physiological stress Erbe et al. (2019) with ecosystem-level consequences when behaviors change over space and time.

This acoustic degradation, exacerbated by the effects of climate change on marine soundscapes and species distributions, creates time-critical monitoring challenges that require robust automated detection systems capable of real-time assessment of species presence, behavioral state changes, and climate-driven population dynamics to inform adaptive conservation interventions. Tittensor et al. (2019); Laidre et al. (2015)

These impacts have motivated concrete mitigation and policy efforts (e.g., quieter ship design, operational routing, and speed management) and targeted recovery planning for St. Lawrence species such as beluga. Our focus in this work is to turn raw hydrophone data into reliable presence signals that support biodiversity protection, monitoring, and adaptation actions in this sensitive region.

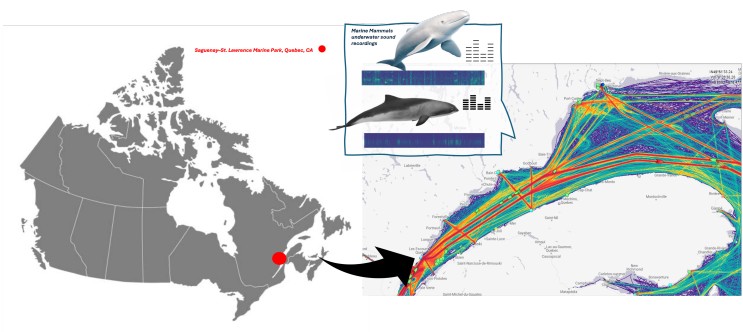

Figure 1: Saguenay–St. Lawrence Marine Park (SSLMP) representation.

**Related Work.** Spectrogram-based approaches have been widely used for underwater mammal sound classification. Cai et al. Cai et al. (2022) proposed a parallel model fusing multi-dimensional acoustic features with transfer learning, while Bach et al. Bach et al. (2023) used STFT-transformed spectrograms combined with a Siamese neural network and VAE for robust classification in noisy environments. Cheng et al. Cheng et al. introduced a hybrid LSTM and multi-scale dilated causal convolution architecture on Mel-spectrograms, showing strong performance across species.

Recent work has also emphasized denoising and mask-guided mechanisms: BirdSoundsDenoising Zhang & Li (2022) inspired our segmentation approach, and methods such as spectrogram-based data augmentation and smoothness-inducing regularization Xu et al. (2024), automatic sound event detection Jiang et al. (2024), wind-robust denoising Juodakis & Marsland (2021), and unsupervised sound separation Denton et al. (2021) highlight the benefit of integrating denoising and attention for improved generalization. These works motivate our Mask-Guided Classification framework for robust underwater bioacoustic analysis.

**Our contributions.** First, we propose an end-to-end multi-modal framework[1] that segments spectrograms to produce pseudo attention masks and fuses mask and spectrogram embeddings to guide denoising and enhance biologically relevant signal recognition. Then we evaluate real-world recordings collected by the Saguenay–St. Lawrence Marine Park Research Station, emphasizing cross-season robustness and per-class precision, with control for empty signals. Finally, we demonstrate that segmentation-driven attention and mid-level fusion improve accuracy, stabilize detection thresholds, and produce robust field-ready representations for underwater bioacoustic monitoring. In addition, we evaluate the generalization capabilities in an out-of-ditribution setting of our proposed mask-guided classifiers through different fusion strategies of acoustic signal representations, and we highlight the superior performance of our approach compared to standard baselines.

## 2 DATASET DESCRIPTION AND PROBLEM SETUP

**Dataset description** We used an exclusive subset of the Saguenay - St. Lawrence Marine Park (SSLMP) monitoring dataset Bernier-Breton (2025), a long-term multimodal collection designed to study the impact of maritime traffic on endangered marine mammals. Data come from two complementary sources: bottom-moored hydrophones (passive acoustic monitoring, PAM) that provide ∼ 1,500 hours of continuous recordings and shore-based surveys (LBS) that provide ∼ 500 hours of visual observations over four years. These data streams are synchronized, producing species-level annotations in Bernier-Breton (2025) for belugas (*Delphinapterus leucas*) and harbour porpoises (*Phocoena phocoena*). Our subset consists of ∼10,000 five-minute segments manually annotated Bernier-Breton (2025) with species presence and sound types (beluga whistles and clicks, 10–100 kHz; porpoise narrowband clicks, 50–150 kHz). The recordings also capture vessel noise and other natural and anthropogenic sounds spanning 10 Hz–150 kHz. The dataset is challenging due to environmental noise, overlapping calls, and domain shifts across seasons, sites, and sensors, making it a unique benchmark for machine learning in underwater bioacoustics.

---

[1]For reproducibility, full details of the framework modeling will be released after the review process.

**Problem setup** We work with a dataset of raw marine acoustic recordings containing vocalizations from multiple species. Our goal is to automatically recognize marine mammal vocalizations in noisy recordings, addressing challenges such as variable signal-to-noise ratios, overlapping calls, and environmental noise. We explore both multi-label and multi-class classification, before introducing attention mask driven framework using spectrogram-based representations of the audio data.

**Formulation** Formally, let $x(t)$ denote a raw acoustic waveform. The signal is first transformed into a spectrogram via a time-frequency representation (STFT). A segmentation model $\mathcal{M}_{\text{seg}}$ predicts a pseudo-attention mask highlighting relevant spectro-temporal regions. Both the spectrogram and the mask are then encoded into embeddings, which are fused to guide denoising and enhance biologically relevant signals. Finally, a classifier $\mathcal{C}$ maps the fused representation to the probabilities of the target class. Formally, the pipeline is:

$$\hat{y} = \mathcal{C}\Big(\text{Fuse}\Big(\mathcal{E}_{\text{spec}}(\mathcal{T}(x(t))), \ \mathcal{E}_{\text{mask}}(\mathcal{M}_{\text{seg}}(\mathcal{T}(x(t))))\Big)\Big), \quad \hat{y} \in \mathbb{R}^K \tag{1}$$

where $\mathcal{T}$ is the STFT, $\mathcal{E}_{\text{spec}}$ and $\mathcal{E}_{\text{mask}}$ are the embedding functions for the spectrogram and mask, respectively, and $\text{Fuse}(\cdot, \cdot)$ denotes the mid-level embedding fusion.

## 3 MASK-GUIDED CLASSIFICATION (MGC) METHOD

**Classification task** The marine mammal acoustic signals were first analyzed by supervised classification in spectrogram representations capturing species-specific signatures. Two paradigms were considered: multi-class classification and multi-label classification. We evaluated convolutional, modern CNN, and transformer-based architectures using standard metrics,

As a transfer learning strategy Bengio et al. (2013), ImageNet normalization was applied to all inputs, given that most models were pretrained on this dataset.

Multi-class classification proved more suitable for our dataset, while noise and artifacts still limit the detection of subtle spectro-temporal patterns (see Fig. 8 and Tab. 4), motivating the denoising framework introduced next.

### 3.1 AUTOMATIC ACOUSTIC DENOISING FRAMEWORK

These difficulties discussed above can be largely attributed to noise that distorts the essential fine-grained temporal and spectral structures. To overcome these challenges, we introduce an automatic acoustic denoising framework designed to preprocess raw audio recordings prior to classification. This framework integrates signal transformation Xu et al. (2024), mask-based denoising Zhang & Li (2022), and classification into a unified pipeline, thus improving robustness by clarifying relevant acoustic patterns through "pseudo-attention" masks and attention mechanisms.

**Framework description** Raw audio signals are first converted into time–frequency representations using the STFT. This operation decomposes the signal into overlapping windows. The resulting spectrograms are then used as the primary visual input for the denoising and classification stages. We apply a denoising methodology inspired by few-shot learning and leveraging the capabilities of models such as DeepLabV3 Chen et al. (2017). A substantial training set is constructed to train a segmentation model that generates "pseudo-attention" masks over spectrograms. These masks are then leveraged in a multi-modal fusion framework, where both the raw spectrogram and its corresponding mask embedding are jointly encoded. The fused representation guides the network to focus on informative regions, effectively denoising the signal and enhancing underwater bioacoustic recognition. This approach is inspired by previous work in the audio denoising domain, notably the study on bird sounds Zhang & Li (2022), which demonstrated the effectiveness of deep visual denoising techniques in improving classification performance.

**Audio transformation and semi-automatic mask labelisation.** The raw audio recordings are first converted to spectrogram representations using standard time-frequency analysis techniques. The spectrograms serve as the primary input for the subsequent denoising and classification stages. Once the spectrogram has been obtained, in order to efficiently annotate large collections, we adopt a semi-automatic labeling approach. First, an initial set of candidate regions is generated using

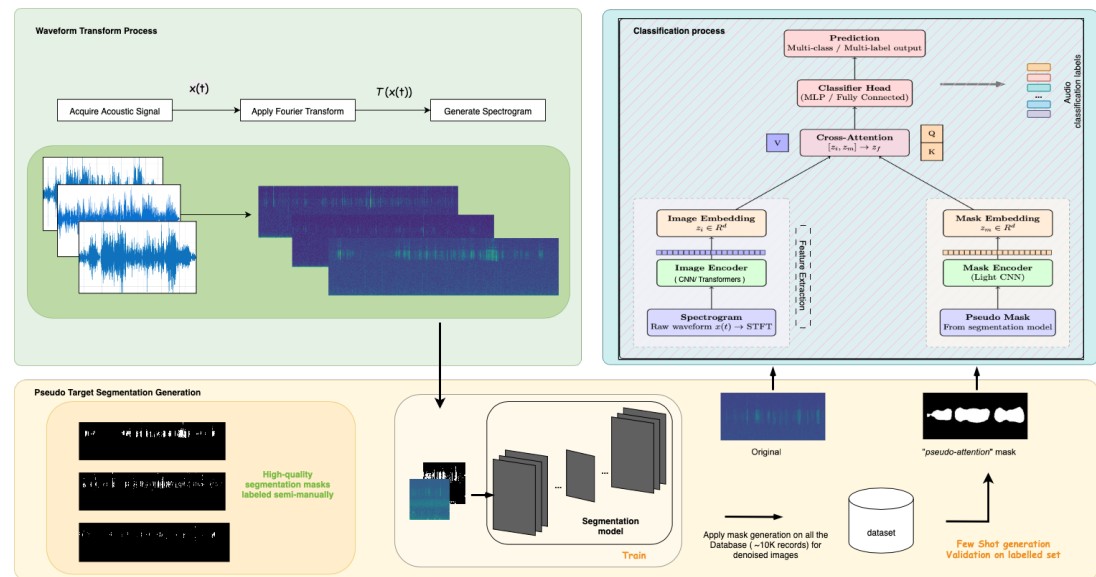

Figure 2: End-to-end framework for automatic denoising and classification from raw audio.

signal processing techniques, such as edge detection and adaptive thresholding, to highlight potential patterns of interest. This allows us to identify and isolate prominent acoustic features. These preliminary masks are then presented to the annotator through an interactive interface, allowing manual refinement and correction, resulting in a high-quality training set (200 images) from which the denoising model can generalize mask predictions across the dataset.

**Few-shot learning for denoising.** Leveraging the high quality mammal sound pattern masks, we train a denoising model using a few-shot learning strategy to generalize from limited annotations. Architectures such as DeepLabV3 capture both fine-grained time–frequency structures and broader contextual patterns to distinguish signal from noise. In addition, we apply image horizontal flip augmentation to double the size of the training dataset. Once trained, the model predicts masks across the full dataset, enabling scalable denoising without exhaustive manual labeling.

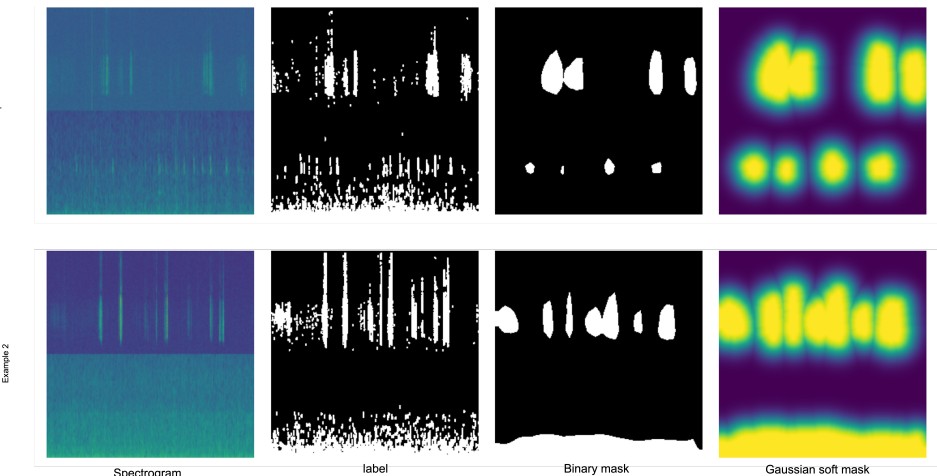

Figure 3: Spectrogram (**left**), high-quality segmentation mask (**middle**), and generated pseudo-attention masks (**2nd and 3rd columns**) for a recording of porpoise clicks (binary and real-valued).

As explained above, the segmentation model allows us to generate *pseudo-attention masks* (third column in fig. 3) for the entire dataset. However, these masks being binary, they capture attention

regions in a strict manner and do not distinguish pixels that are close to regions of interest, which may contain relevant information, from those that are far from the signal. The goal is therefore to transform these binary masks to encode the spatial proximity of information.

Let a binary mask be denoted as $M \in \{0,1\}^{H \times W}$, where $M(x,y) = 1$ indicates a region of interest and $M(x,y) = 0$ the background. For each background pixel, we compute the Euclidean distance to the nearest pixel in the region of interest:

$$d(x,y) = \min_{(x',y') \in \Omega} \|(x,y) - (x',y')\|_2, \quad \Omega = \{(x',y') \mid M(x',y') = 1\}. \tag{2}$$

This distance map $d(x,y)$ is then transformed into a *soft mask* $S(x,y)$ using a decreasing Gaussian function:

$$S(x,y) = \exp\left(-\frac{d(x,y)^2}{2\sigma^2}\right), \tag{3}$$

where $\sigma$ controls the spatial decay. Pixels inside the original mask region retain values close to 1, while pixels farther from the region of interest gradually decrease in value.

This transformation thus relaxes the binary mask into a continuous mask $S(x,y) \in (0,1]$ (see fig. 3), encoding not only the strict presence of the signal but also its spatial proximity, which is particularly useful for attention-based tasks and for weighting relevant information in our spectrograms.

**Mask-guided multimodal model for classification.** After training our segmentation model on spectrograms, we obtain pseudo-attention masks that highlight regions most likely to contain relevant acoustic events. Following the approach of Zhang and Li Zhang & Li (2022), we use segmentation to prevent the model from considering additional noise in the signal. Intuitively, the mask acts as a form of attention-based denoising: it emphasizes salient regions of the spectrogram while suppressing background noise and irrelevant structures (see fig. 3. Concretely, we design a multimodal fusion framework with two parallel encoding branches: **Spectrogram encoder**, a ResNet50 or audio transformer backbone processes the raw spectrogram into a high-level representation. **Mask encoder**, a lightweight CNN encodes the corresponding segmentation mask into a compact embedding. Both embeddings are projected into a common latent space and then fused at an intermediate stage (mid-fusion). Fusion can be realized either by simple concatenation or through a cross-modal attention mechanism, where the spectrogram embedding serves as the query and the mask embedding provides keys and values. This enables the network to adaptively weigh spectro-temporal regions conditioned on the mask.Then, the fused representation is passed to a classification head, producing multi-class predictions. This design preserves a residual path from the spectrogram encoder to the classifier, ensuring that the system does not overly rely on potentially noisy masks while still exploiting their guidance signal. In doing so, we approximate the role of human attention in auditory scene analysis: focusing on the most informative patterns while filtering out distracting background components.

## 4 RESULTS

### 4.1 EVALUATION METRIC

In this work, we evaluate the models using a metric that we refer to as *joint accuracy*. The task is formulated as a multi-label classification problem with three possible classes: (i) presence of whistles, (ii) presence of beluga clicks, and (iii) presence of porpoise clicks. For evaluation, this problem is reformulated as a multi-class classification task by considering all possible combinations of these three classes, including the absence of all. Each observation can therefore belong to one of eight categories, depending on which combination of species is present.

We represent the labels as a binary vector

$$\mathbf{y} = (y_{\text{whistle}}, y_{\text{beluga}}, y_{\text{porpoise}}) \in \{0,1\}^3, \tag{4}$$

where each component indicates the presence (1) or absence (0) of the corresponding signal. The output space is therefore $\mathcal{Y} = \{0,1\}^3$, which contains $2^3 = 8$ possible combinations (including the absence of all signals).

Given a dataset of $N$ observations with true labels $\{\mathbf{y}^{(i)}\}_{i=1}^{N}$ and corresponding predictions $\{\hat{\mathbf{y}}^{(i)}\}_{i=1}^{N}$, the *joint accuracy* is defined as

$$\text{Joint Accuracy} = \frac{1}{N} \sum_{i=1}^{N} \mathbf{1}\left(\hat{\mathbf{y}}^{(i)} = \mathbf{y}^{(i)}\right), \tag{5}$$

where $\mathbf{1}(\cdot)$ is the indicator function, equal to 1 if the condition holds and 0 otherwise.

The *joint accuracy* measures whether the model correctly predicts all classes simultaneously for a given observation. This choice of metric is motivated by the practical needs of researchers, who are often interested in detecting the simultaneous presence of different species or in studying their interactions and communication patterns. Therefore, it is crucial to evaluate models on their ability to capture all relevant acoustic events at once, rather than focusing on each class in isolation.

### 4.2 DENOISING PROCESS FOR MARINE MAMMALS RECOGNITION

To evaluate the contribution of the proposed multimodal denoising framework, we compared it with standard image-only classification models trained on the same data set. Table 1 reports the joint accuracy and macro-F1 in ResNet50 He et al. (2015), ConvNeXt Liu et al. (2022), ViT Dosovitskiy et al. (2021); Touvron et al. (2020), and our cross-attention fusion model using generated or high-quality (HQ) segmentation masks.

In general, the results show that the multimodal approach substantially outperforms all baselines. Although ViT already provides strong performance among unimodal models (78.8% accuracy), suggesting that attention mechanisms are better suited to model long-range temporal and spectral dependencies, the use of generated masks with cross-attention further improves the results to 83.7%.

| Model | Accuracy | F1 macro |
|-------|----------|----------|
| ResNet50 | 0.588 | 0.562 |
| ConvNeXt | 0.625 | 0.591 |
| ViT | 0.788 | 0.787 |
| Multimodal (Gen. masks) | 0.837 | 0.816 |
| Multimodal (HQ masks) | **0.897** | **0.890** |

Table 1: Comparison of baseline image-only models and the proposed multimodal approach with cross-attention using either generated or a **subset** with high-quality masks.

The best performance is obtained with HQ masks (89.7% accuracy, 89.0% macro-F1), highlighting the benefit of leveraging accurate structural priors for denoising. This indicates that cross-attention enables the model to effectively exploit mask information to focus on relevant acoustic structures, and helps for the robustness of the classification.

### 4.3 ABLATION STUDIES

#### 4.3.1 FUSION METHODS

| Fus. strategy | High-Quality Masks | | | | Generated Masks | | | |
|---------------|------------|-----------|----------|----------|------------|-----------|----------|----------|
| | Train Loss | Train Acc. | Val. Loss | Val. Acc. | Train Loss | Train Acc. | Val. Loss | Val. Acc. |
| Concat | 0.370 | 0.887 | 0.559 | 0.762 | 0.365 | 0.877 | 0.678 | 0.825 |
| Gated | 0.401 | 0.868 | 0.792 | 0.713 | 0.472 | 0.833 | 0.857 | 0.762 |
| xAttn | 0.253 | 0.912 | 0.406 | **0.900** | 0.427 | 0.843 | 0.695 | **0.838** |

Table 2: Comparison of mid-fusion strategies on the validation set using either high-quality (HQ) or generated (Gen.) masks. Cross-attention consistently achieves the best validation accuracy. (Training with RTX A100 GPU $\sim$ 15min per method)

We conducted an ablation study on the fusion strategy, comparing simple concatenation, gated residual fusion, and cross-attention; the results (Table 2) show that cross-attention achieves the best validation accuracy. These results suggest that, while simple and gated fusion capture some complementary information between the image and the mask but is more efficient with generated masks, introducing cross-attention enables more effective interaction between modalities.

### 4.3.2 Gaussian soft mask

As described earlier, we introduce an improved version of the binary mask. When comparing the results obtained with binary masks and enhanced soft masks (see bottom of table 3 and table 2), we observe that the effect of the mask transformation is not consistent across all fusion strategies. For example, with the binary mask, Concat and X-Attn achieve accuracies of 0.825 and 0.838 respectively, while Gated Fusion reaches 0.762. After introducing the enhanced mask, X-Attn improves further to 0.863, but both Gated Fusion and Concat Fusion drop to 0.713 and 0.725, respectively. This suggests that the enhanced mask is not universally beneficial across fusion methods. In the case of Gated Fusion, the model is designed to explicitly learn how much information to pass from the mask to the feature representation. Replacing the binary mask with a continuous one may reduce the gating mechanism's effectiveness, as the mask itself already embeds a notion of graded relevance, leading to redundancy or even conflicting signals during training. Similarly, Concat Fusion may suffer because the continuous mask introduces smoother variations that are harder to exploit when features are concatenated directly without explicit weighting.

In contrast, X-Attn benefits from the enhanced mask because the attention mechanism is naturally suited to leverage graded spatial information. The soft mask provides a richer contextual signal for attention weights, allowing the model to focus not only on the exact region of interest but also on its spatial neighborhood. This synergy explains the consistent improvement observed for X-Attn, while the other methods, which rely more heavily on the explicit binary guidance, show a decline in performance.

## 4.4 Out-of-Distribution Generalization Performance

We evaluate the performance of our models on data drawn from a distribution different from the training set (see fig. 5 and 3). Our framework relies on specific transformations of the raw acoustic recordings to generate spectrograms. These transformations are parameterized according to the frequency range of interest, intensity scaling, and other signal-processing parameters. When applying an alternative transformation, as in the case of the additional 10,000 evaluation images, the task becomes an out-of-distribution (OOD) scenario.

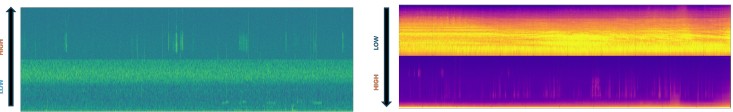

Figure 4: Representation of origin sample on which the model is trained **(left)**, and out-of-distribution sample from a different signal transformation (**(right)**

In this OOD setting, we find that performance remains relatively stable for ResNet50, ConvNeXt, and for the Mask-Guided Classification (MGC) models with gated and concat fusion. These models, which were the least accurate on the original distribution, do not suffer from degradation and thus maintain comparable accuracy across both datasets. The slight improvement of this models on the OOD dataset may be explained by the fact that the alternative transformation better highlights the acoustic patterns of marine mammal vocalizations.

In contrast, the strongest models on the original set (ViT-B/16 and MGC with cross-attention) lose performance and exhibit a marked accuracy drop under OOD evaluation.

This suggests that high-capacity models, which leverage the richness of the original transformation and attention masks, are more vulnerable to distributional shifts. Their learned representations appear over-specialized to the training distribution, leading to weaker robustness when exposed to unseen transformations. Conversely, simpler architectures and less sophisticated fusion strategies (gated, concat) may capture more generic patterns, which, although less optimal on the training set, yield better stability in generalization.

Finally, the observation that concat and gated fusion yield nearly identical results on the 10k OOD dataset supports the hypothesis that, once sufficient training data are available, even simple fusion mechanisms can allow the model to effectively exploit the relationship between the original spectrogram and the pseudo-attention mask.

Table 3: Comparison of classification performance between baseline models and our mask-guided approach, evaluated on in-distribution (Origin) and out-of-distribution (OOD) datasets.

| Method | Training | Evaluation | Accuracy (Val) |
|---|---|---|---|
| **Baseline (full training on 10k samples, standard models)** | | | |
| ResNet50 | Origin | OOD | 0.638 |
| ConvNeXt-Tiny | Origin | OOD | 0.675 |
| ViT-B/16 | Origin | OOD | 0.616 |
| **Segmentation mask (train on Origin, test on OOD)** | | | |
| X-Attn | Origin | OOD | 0.750 |
| Gated Fusion | Origin | OOD | 0.743 |
| Concat Fusion | Origin | OOD | **0.765** |
| **Enhanced mask (Gaussian soft mask & direct real mask)** | | | |
| X-Attn | Origin | Origin | **0.863** |
| Gated Fusion | Origin | Origin | 0.713 |
| Concat Fusion | Origin | Origin | 0.725 |
| X-Attn | Origin | OOD | 0.742 |
| Gated Fusion | Origin | OOD | 0.735 |
| Concat Fusion | Origin | OOD | 0.746 |

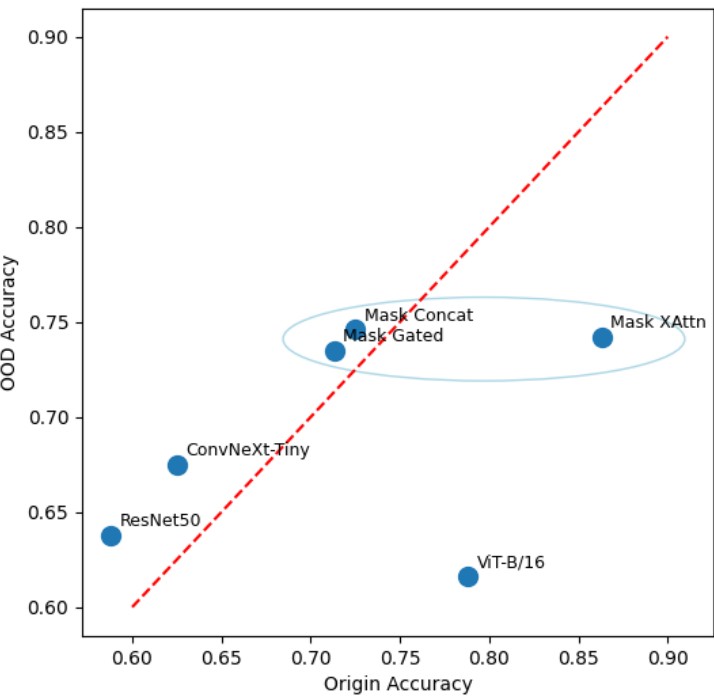

Figure 5: Comparison of the **joint accuracy** of the different models on the original data and on the out-of-distribution data. Models from the mask-guided methodology are highlighted in blue.

In all cases, the use of our Mask-Guided Classification (MGC) framework substantially improves performance compared to baseline models, both on the original distribution and under OOD evaluation.

## 5 DISCUSSION

While our framework demonstrates promising results, it inherits some limitations from the signal transformation choices. In particular, STFT can introduce resolution trade-offs and information loss, which may restrict the model's ability to fully capture the complexity of marine mammal vocalizations. Moreover, our study did not incorporate explicit uncertainty quantification, an aspect that is increasingly important for trustworthy machine learning in ecological monitoring. Future work will address these issues by exploring alternative time–frequency representations, improving attention mechanisms, and integrating methods to quantify predictive uncertainty, thus making the framework more robust and reliable for scientific and conservation-oriented applications.

## 6 CONCLUSION

We introduced a segmentation-guided multimodal framework that consistently improves recognition of marine mammal vocalizations under real-world noise and overlap. By fusing spectrogram and mask embeddings via mid-level cross-attention, the method produces reliable and interpretable presence signals that align with independent visual surveys. This establishes a principled route to scientific inference from raw acoustic signals, with immediate relevance to ecology and broader acoustic sensing problems.

One of the aims of our experiments was to rigorously assess the robustness and generalization capabilities of various models under distributional shifts. We compared performance on original versus out-of-distribution (OOD) data generated through alternative spectrogram transformations, and investigated how different architectures respond to such OOD scenarios.

Our results indicate that, the Mask-Guided Classification (MGC) framework consistently support generalization, particularly when paired with attention-based fusion mechanisms, highlighting the benefits of incorporating spatially-informed mask guidance into the classification process.

Overall, this work demonstrates that carefully designed mask-guided approaches, combined with appropriate fusion strategies, can significantly enhance the resilience of bioacoustic classifiers to distributional shifts, providing a practical framework for robust underwater acoustic monitoring. Deep learning models can extract reliable presence signals that directly support species monitoring and conservation, illustrating how these techniques can be effectively harnessed for scientific and climate-relevant ocean studies.

To facilitate reproducibility and further research, all code associated with this work are made publicly available as open-source resources on GitHub.

### ACKNOWLEDGMENTS

This work was conducted as part of a research experiment at MILA - Quebec Artificial Intelligence Institute and The Research Chair of AI for Suply Chains at University of Quebec at Rimouski (UQAR). We gratefully acknowledge the financial and technical support provided by MILA, GERAD, XST and the Chair which made this research possible. We also thank the Saguenay–St. Lawrence Marine Park monitoring program for providing access to this high-quality database.

## REPRODUCIBILITY STATEMENT.

We are committed to ensuring the reproducibility of our results. While data are not yet publicly available, we are making our best efforts to share them in the near future to contribute to the research community. All architectural details, framework steps, and experimental protocols are described thoroughly in the main text, appendix, and supplementary materials, providing sufficient information to reproduce the main experimental results once the resources are released. In particular, we provide detailed descriptions of the Mask-Guided Classification framework, the fusion strategies, and the training procedures. Open access to data will be provided as soon as possible to facilitate faithful reproduction of our experiments.

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

# A   APPENDIX

You may include other additional sections here.

## A.1   SAGUENAY–ST. LAWRENCE MARINE PARK REPRESENTATION

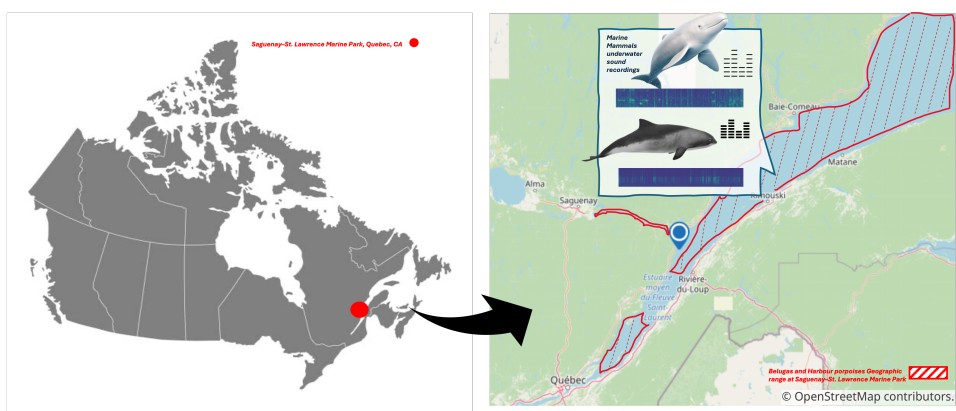

Figure 6: Saguenay–St. Lawrence Marine Park (SSLMP) representation.

## A.2   ARCHITECTURE DETAILS

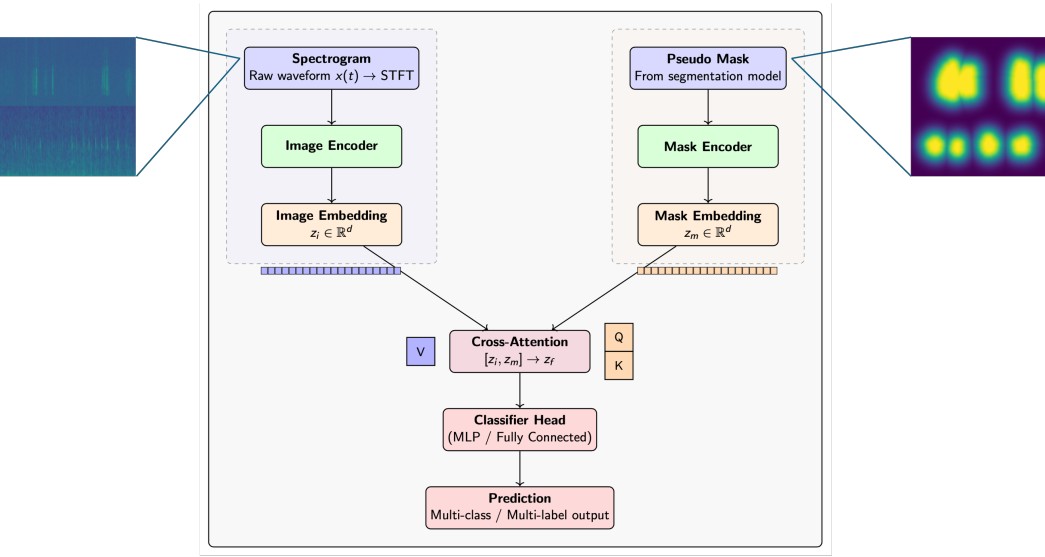

Figure 7: Architecture of the proposed model with two encoding branches and mid-fusion by cross-attention

## A.3 CLASSIFICATION TASK

Table 4: Performance comparison between multi-label and multi-class training approaches before multi-modal approach. *For multiclass (one label per sample): hamming loss is the average number of incorrect predictions per sample. For multilabel (multiple labels per sample): it is the average number of label errors per sample, divided by the number of labels. This metric is not comparable inter training method*

| Metric | ConvNeXt-Tiny | | ResNet50 | | Deit-Distilled | |
|---|---|---|---|---|---|---|
| | Multi-Label | Multi-Class | Multi-Label | Multi-Class | Multi-Label | Multi-Class |
| **Hamming Loss** | 0.1693 | 0.3310 | 0.1206 | 0.3466 | 0.1427 | 0.3674 |
| **Perfect Accuracy** | 58.17% | **66.90**% | 66.34% | 65.34% | 62.45% | 63.26% |
| **Whistle** | | | | | | |
| Precision | **0.806** | 0.61 | 0.745 | 0.60 | 0.730 | 0.64 |
| Recall | **0.891** | 0.82 | 0.816 | 0.77 | 0.745 | 0.71 |
| F1-Score | **0.847** | 0.70 | 0.779 | 0.68 | 0.737 | 0.67 |
| **Beluga Click** | | | | | | |
| Precision | 0.672 | 0.68 | **0.968** | 0.63 | 0.926 | 0.71 |
| Recall | **0.996** | 0.77 | 0.921 | 0.57 | 0.939 | 0.50 |
| F1-Score | 0.802 | 0.72 | **0.944** | 0.60 | 0.932 | 0.59 |
| **Porpoise Click** | | | | | | |
| Precision | 0.868 | 0.68 | **0.966** | 0.67 | 0.925 | 0.69 |
| Recall | 0.985 | 0.73 | 0.957 | 0.53 | **0.979** | 0.48 |
| F1-Score | 0.922 | 0.71 | **0.961** | 0.59 | 0.951 | 0.57 |

(a) Multi-labels trained classifiers performances.

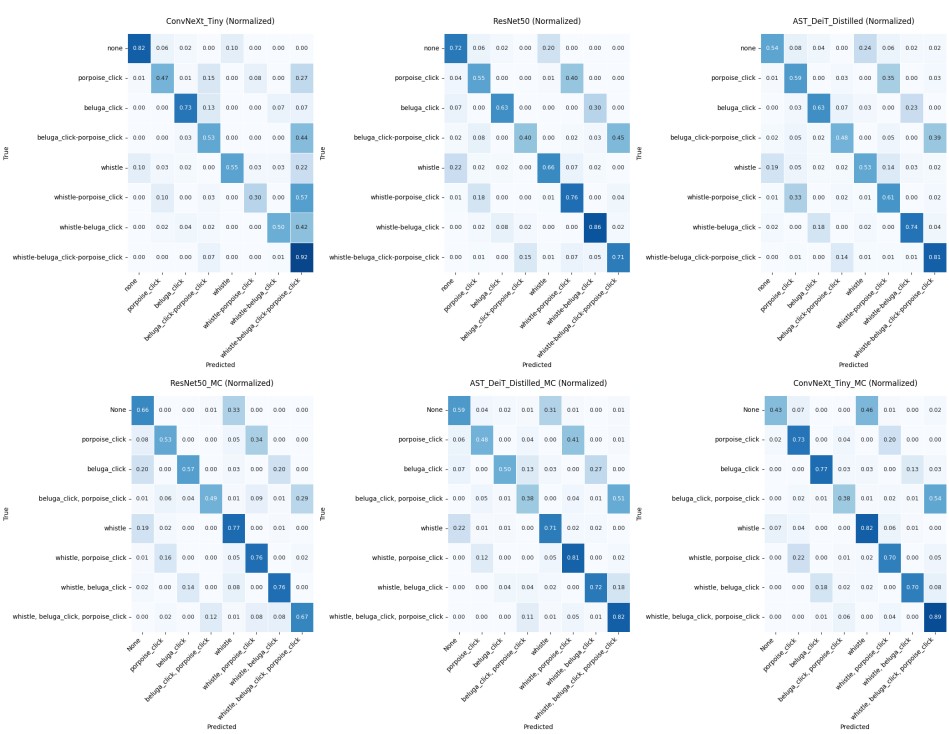

(b) Multi-classes trained classifiers performances.

Figure 8: Comparison of classifiers trained with multi-labels (top row) vs. multi-classes approaches(bottom row) before integration of attention masks. Values are normalized by the size of the test set and represent the percentage of well classified labels.

