# OpenReview forum: "Multi-Representation Attention Framework for Underwater Bioacoustic Denoising and Recognition"
_ICLR.cc/2026/Conference — Submitted to ICLR 2026_

### Official Review · Reviewer_iuEZ · 2025-10-29

**Soundness:** 2
**Presentation:** 2
**Contribution:** 2
**Rating:** 4
**Confidence:** 3

**Summary:**

This paper introduces a multi-step, attention-guided framework that first segments spectrograms to generate soft masks of biologically relevant energy and then fuses these masks with the raw inputs for multi-band, denoised classification in marine mammals automated monitoring scenarios.

**Strengths:**

This work proposes a new processing flow that combines multiple existing works and applies it in marine mammals automated monitoring scenarios.

**Weaknesses:**

1. In the context of marine mammals automated monitoring scenarios, the motivation for integrating spectrograms and masks has not been clearly explained, as it seems to simply copy existing technology routes without relevant ablation experiments to demonstrate the contributions of each branch.
2. Using multiple classifications instead of multiple labels is feasible when there are few categories, but as the number of categories increases, the approach proposed in the paper can lead to an exponential growth in the number of model classifications, which limits its application to other datasets and tasks.
3. The model parameter settings, training parameter settings, and dataset preprocessing parameter settings for the experiment are all opaque. The models compared in the experiment did not include the most recent models.
4. Using real datasets as out of distribution datasets is more convincing, and the parameters involved in generating out of distribution datasets are not transparent.
5. There are some writing irregularities in the appendix, such as including "You may include other additional sections here." which is irrelevant to this work.
6. The task settings in the experiment are simple and cannot reflect or demonstrate the performance of the model in high noise environments.

**Questions:**

1. Has this work been compared with the recent sate-of-the-art (SOTA) models in this field?
2. Is there a detailed explanation of the model's denoising ability and recognition ability in high noise environments through control experiments?
3. Is fusing spectrograms and masks the best solution? Do you have ablation experiments?
4. Can experiments be conducted on other datasets with more category recognition to demonstrate the model's ability?
5. Can real datasets be used as out of distribution data to demonstrate the model's generalization ability?

---

### Official Review · Reviewer_7NnC · 2025-10-29

**Soundness:** 1
**Presentation:** 1
**Contribution:** 2
**Rating:** 2
**Confidence:** 4

**Summary:**

This paper’s main contribution is the introduction of Mask-Guided Classification (MGC), a novel method for classification in (marine mammal) bioacoustics. The approach first generates segmentation masks from spectrograms, which are then relaxed into soft, probabilistic guides. These masks are fused with spectrogram embeddings at a mid-level stage using one of several strategies (concatenation, gating, or cross-attention). The results show that MGC, particularly when paired with a cross-attention fusion mechanism, significantly improves in-distribution classification performance over standard backbone models.

**Strengths:**

- Practical, domain-aware design: The idea of using segmentation-derived soft masks is a good idea for bioacoustics. It provides a way to guide the model's attention towards biologically relevant signal components while preserving global context.
- Strong in-distribution performance: The paper demonstrates clear performance gains on in-distribution data, with the cross-attention fusion strategy showing the most substantial improvements.
- Ablations: The authors provide ablations that compare different fusion strategies and analyze the specific benefits of soft, graded masks over binary ones. This strengthens the paper's conclusions regarding why the cross-attention mechanism is particularly effective.

**Weaknesses:**

**Presentation and related work**

- **Insufficient related work:** The literature review is very limited (only a handful of cited papers), failing to properly position the work within the current landscape of bioacoustic machine learning. Especially other marine models (e.g., Surfperch) and other bioacoustic tasks like avian bioacoustics provides a great body of literature that would help positioning this paper.
- **Poor citation practices and formatting:** The paper contains numerous errors, including repeated citations (e.g., Cai et al. Cai et al. in L.69, Bach et al. Bach etal. in L.70), citation of a master's thesis (and very limited other peer-reviewed papers), placeholder text (L596: “You may include other additional sections here”, and citations with only an author name in L513 ). A claim attributed to a paper by Denton et al. in L.78 also appears to be incorrect (attention is not used in this context).
- **Unreadable figures:** The figures, especially Figure 1 and Figure 2, are not readable due to low resolution or small text. Figure 1 is not referenced in the text and lacks an informative caption, making it hard to understand its purpose (even though you can guess it).


**Superficial evaluation and missing baselines (bigges problem):**

- **Weak OOD definition:** The out-of-distribution (OOD) evaluation is superficial. Inducing a domain shift by only altering spectrogram parameters does not reflect real-world challenges like different recording sites, sensor types, seasons, or acoustic habitats. While it is an interesting take of the problem, this definition limits the credibility of any generalization claims.
- **Absence of SOTA baselines or other datasets:** The evaluation is missing comparisons to established, SOTA animal audio classification models (e.g., Perch, or Bird-MAE) or other datasets in the bioacoustic domain (e.g., BirdSet) to test the validity of the method (at least some small scale studies). Additionally any AudioSet-pretrained models are missing (why would I even choose ImageNet-pretrained models if audio models are available?) Without these baselines, the reported performance gains are very difficult to contextualize.
- **Non-standard metrics and lack of rigor:** The paper does not report standard multi-label classification metrics (e.g., mAP, AUROC), which limits comparability with other literature (while there is the motivation for the other metric, it would still be nice to include them for comparison). Furthermore, the absence of confidence intervals or results from repeated seeds questions the statistical robustness of the findings.

**Lack of methodological clarity and justification:**

- **Ambiguous model architecture:** It is unclear what specific backbone architecture (e.g., ResNet, ViT) was used in the proposed setup. While there are vague information throughout the paper, it did not become clear to the reader.
- **Unjustified pre-training strategy:** The choice of ImageNet pre-training is poorly motivated. Given the availability of strong audio-pretrained models (e.g., on AudioSet), the decision to use a model pretrained on natural images for an audio task is non-standard and requires a strong justification or a comparative experiment. It is pretty clear that ImageNet-pretrained models do not perform that well in an acoustic environment
- **Unclear experimental details:** Some details are missing or are ambiguous. For example, Table 3's distinction between "full training on 10k samples" and training on "Origin" is not clear. There is also no mention of a hyperparameter study.


**Reproducibility:**

- While the authors say that they will release the code in the future, I would expect an anonymized version of the code during the review phase. The paper provides neither the source code for the method nor a concrete plan for the public release of the new dataset.

**Questions:**

1. To validate the practical utility of MGC, could you evaluate its OOD generalization on more realistic domain shifts in bioacoustics (e.g., on a BirdSet task)? Furthermore, how does your model compare against established, domain-specific bioacoustic baselines like Perch, SurfPerch or Bird-MAE etc.?
2. Could you clarify the specific backbone architecture used for your experiments and provide a strong justification for using ImageNet pre-training over more standard audio-centric pre-training? Additionally, what is your concrete plan for releasing the source code and the newly collected dataset (e.g. where and how) to ensure reproducibility?

---

### Official Review · Reviewer_ZB1o · 2025-10-30

**Soundness:** 2
**Presentation:** 3
**Contribution:** 2
**Rating:** 4
**Confidence:** 3

**Summary:**

The paper presents a method to improve classification performance by fusing spectrogram inputs with generated soft-masks to denoise underwater bioacoustic signals. The proposed mask-guided classification method consists of a trained segmentation model that produces soft masks, which are fused mid-way into a classification model via cross-attention. The model is evaluated on underwater bioacoustic dataset of marine mammals with three distinct classes. The model is compared to image-based baselines without audio-specific architectures or pretraining and shows significant improvements.

**Strengths:**

- Presented framework is novel and presented results show significant improvements over baselines.
- The paper is generally well-written and easy to follow.

**Weaknesses:**

- The experimental design is hard to follow, it is unclear for me:
    - What data is exactly used for which parts of the training and how it is split?
    - Which architecture / pretraining checkpoint is used in the "Multimodal" model?
- The presented framework is evaluated on a new dataset only making it hard to assess the significance of the results.
- The presented baselines are from the image domain only, no audio or bioacoustic baselines are included.
- Formatting glitch in l.250, l.513

**Questions:**

- How are hyperparameters selected?
- How many seeds were used for the experiments? What are the standard deviations?
- Would it be possible to test OOD generalization on a totaly different dataset, e.g. Watkins marine mammal sounds?

---

### Official Review · Reviewer_pyrD · 2025-11-02

**Soundness:** 2
**Presentation:** 2
**Contribution:** 2
**Rating:** 4
**Confidence:** 4

**Summary:**

The paper proposes an end-to-end multi-modal framework to represent bioacustic signals for monitoring. The study is done with data from the St. Lawrence Estuary habitat.

**Strengths:**

The problem statement is interesting and attractive setting for studying practical applications of machine learning.

The study can be of interest for the community working in bioacustics.

**Weaknesses:**

The technical contribution is very small.

Some of the formulation is non-standard. For example: equations 2 to 3 seem to be describing a convolution between the binary mask an a gaussian kernel (normalized so that the sum is equal to 1), but it is done in a complicated way.

Equation 5 is not needed as this is a very standard error metric.

**Questions:**

Could you better describe the difference between multi-class classification and multi-label classification?

Are equations 2 and 3 just a convolution?

In the case of audio signals, does it make sense to augment the dataset doing an image horizontal flip augmentation? Doesn’t this introduce non-natural sounds?

---

### Meta-Review · Area_Chair_3fRW · 2026-01-04

**Summary:**

The reviewers agree that the problem setting is interesting, although largely to a narrow audience, but also as a good application of machine learning. That said, there are several issues related to the quality of submission for the current venue:
1. Overall contribution seems small, given masking / masking based robustness solutions are widely used in speech and image processing tasks.
2. Task is too narrow, so it’s unclear how general the presented framework is for other related tasks.
3. Missing relevant baselines (e.g., using audio instead of just the image).
4. General issues with clarity of presentation, and lack of rigor when presenting results.

**Reviewer Concerns:**

The authors did not submit a rebuttal.

**Reviewer Scores:**

n.a.

---

### Decision · Program_Chairs · 2026-01-26

Reject